# Entanglement of orbital angular momentum in non-sequential double ionization

Andrew S. Maxwell [1,2] ✉, Lars Bojer Madsen [2] & Maciej Lewenstein [1,3]

Entanglement has a capacity to enhance imaging procedures, but this remains unexplored for attosecond imaging. Here, we elucidate that possibility, addressing orbital angular momentum (OAM) entanglement in ultrafast processes. In the correlated process non-sequential double ionization (NSDI) we demonstrate robust photoelectron entanglement. In contrast to commonly considered continuous variables, the discrete OAM allows for a simpler interpretation, computation, and measurement of entanglement. The logarithmic negativity reveals that the entanglement is robust to incoherence and an entanglement witness minimizes the number of measurements to detect the entanglement, both quantities are related to OAM coherence terms. We quantify the entanglement for a range of targets and field parameters to find the most entangled photoelectron pairs. This methodology provides a general way to use OAM to quantify and measure entanglement, well-suited to attosecond processes, and can be exploited to enhance imaging capabilities through correlated measurements, or for generation of OAM-entangled electrons.

One of the most notorious departures from classical physics is quantum entanglement, a subtle combination of classical correlation and quantum superposition, which both caused a seismic shift in how the world is viewed, and also provides a resource for quantum computation and metrology[1–3], including the possibility of entanglement enhanced imaging[4,5], with much interest in atomic and molecular physics, e.g., properties of entangled photons from such systems[6–8]. However, the potential for entanglement to optimize attosecond ($10^{-18}$s) imaging processes is unexplored, and therefore the role and physical insight afforded by entanglement for such processes remains unclear.

Attosecond and strong-field physics deal with processes in matter on the scale of attoseconds[9,10]. The promise of resolving atomic and molecular electron dynamics on its natural timescale has led to the development of a host of imaging procedures boasting attosecond time resolution; including strong-field initiated methods like high-order high harmonic spectroscopy[11], laser-induced electron diffraction[12,13], photoelectron holography[14,15] and attosecond pump-probe techniques like attosecond streaking[16,17] and reconstruction of attosecond harmonic beating by interference of two-photon transitions[18,19]. Although most of these imaging protocols depend on

quantum processes, none explicitly exploit entanglement. Recently, the quantum nature of attosecond processes was exhibited through the generation of non-classical states of light in the laser driving field[20,21].

From as early as 1994[22], there has been a growing interest in the role of entanglement in attosecond processes. Many studies have focused on entanglement between photoelectrons and ions[23–31], an essential part of understanding decoherence[32], while other studies have focused on electron–electron entanglement[33–35]. However, the majority of studies involve the calculation of a continuous variable density matrix (exceptions include studies focused on entanglement involving discrete vibrational states, see, e.g., Refs. [28,30,31]) and entanglement measures such as the purity[26,29,30,34,35] or von Neumann entropy[26,27,29,35]. These quantities, when derived from continuous variables, have some drawbacks: (i) They are challenging to compute, and often approximations must be imposed, or in limited cases analytical approximations can be found[36]. Furthermore, these methods are restricted to pure states, while in strong-field experiments we must always consider mixed states from incoherence averaging. (ii) The physical interpretation of these quantities is difficult, and may not

[1]ICFO-Institut de Ciencies Fotoniques, The Barcelona Institute of Science and Technology, Av. Carl Friedrich Gauss 3, 08860 Castelldefels, Barcelona, Spain.
[2]Department of Physics and Astronomy, Aarhus University, DK-8000 Aarhus C, Denmark. [3]ICREA, Pg. Lluís Companys 23, 08010 Barcelona, Spain.
✉e-mail: andrew.maxwell@phys.au.dk

improve understanding of the process. (iii) Direct experimental evidence of the entanglement is often practically impossible, requiring the measurement of incompatible observables, such as momentum and position.

One solution to these difficulties is to use quantized observables. All free particles have such a quantum observable in the form of orbital angular momentum (OAM)[37,38]. Photons carrying OAM have received significant attention in producing extreme ultraviolet (XUV) high-order harmonics with OAM, see e.g., Refs. [39–42]. Strong-field studies on OAM in photoelectrons include exploiting OAM in rescattering electrons to probe bound state structures[43], and recent work[44–47], providing insight into the role of OAM for circularly polarized fields and conservation laws for OAM in strong-field ionization[48]. Measurement of OAM is a rapidly expanding field, with a host of techniques becoming available for electron beams[49–56]. Conservation between the initial quantum magnetic number and final OAM, which occurs for systems with rotational symmetry around the quantization axis, may be exploited in strongly-correlated two-electron processes, where entanglement could allow for enhanced photoelectron imaging.

Non-sequential double ionization (NSDI) is a highly correlated two-electron ionization process, the details of which are depicted in Fig. 1. Despite strong electron–electron correlation and rescattering being confirmed in NSDI as early as 2000[57,58], there has been little focus on the quantum entanglement between the two electrons. This is primarily for the following reasons, (i) classical models have been very successful in modelling NSDI[59], (ii) early work suggested entanglement would not play a decisive role. These studies focused on the momentum coordinate parallel to the laser field and found a small degree of quantum correlation[33], later it was shown classical correlation was sufficient for field intensities greater than $10^{14}$ W/cm²[60]. (iii) Furthermore, computation of NSDI is a very arduous task and computation of continuous variable entanglement measures is even more difficult.

In this work we address (ii) and (iii), by exploiting the quantized nature of OAM to clearly demonstrate entanglement in NSDI, which we show may occur most easily through the RESI pathway. The use of a quantized degree of freedom enables a simple analysis through the logarithmic negativity[61] and entanglement witnesses[62–64], which enables the inclusion of incoherent effects, as well as a search over a wide range of parameters. The interplay of channels of excitation allows photoelectrons to approach maximally entangled states for some final momenta, which could be investigated as a source of OAM entangled electrons. We show that the entanglement exhibited is robust to incoherent averaging of laser intensities over the focal volume. By decomposing an entanglement witness, we strongly reduce the difficulty of detecting entanglement, by avoiding full

tomographic measurements. Furthermore, the OAM entanglement could be used to perform correlated OAM and momentum measurement on the two electrons. Thus, paving the way for a unique kind of entanglement enhanced attosecond imaging technique.

## Results

### Orbital angular momentum in non-sequential double ionization

A pictorial depiction of non-sequential double ionization (NSDI) is given is Fig. 1. The process follows the three-step mechanism [panel (b)][65]: (i) The first electron is removed from the two-electron ground state $|0\rangle$ by the laser via tunnel ionization into the state $|\tilde{\mathbf{p}}, 0\rangle$ (one electron in the continuum and the other in its ground state). (ii) The continuum electron subsequently undergoes a laser driven recollision with its parent ion. (iii) The energy imparted by the collision allows for two pathways, a second electron is directly ionized in the electron-impact ionization (EI) mechanism, or the second electron is excited, resulting in the state $|\tilde{\mathbf{p}}, \eta\rangle$—here $\eta$ is used to label the excited state—and the second electron subsequently ionizes due to the laser field in the recollision with subsequent ionization (RESI) mechanism. In both cases, the final state of the photoelectrons is $|\tilde{\mathbf{p}}, \tilde{\mathbf{p}}'\rangle$, the two electron continuum state. For more information on this notation, see ref. [66], while for reviews of NSDI and these mechanisms, see refs. [67,68]. The laser field is polarized in the $z$ direction, in the same direction as the total OAM operator $\hat{L}_\parallel$, so the laser field cannot change the total OAM since $[\hat{L}_\parallel, \hat{H}(t)] = 0$, where $\hat{H}(t)$ is the total Hamiltonian of the system[44,48].

We expand the NSDI two-electron continuum wave function in a basis of electron vortex states, the one-electron vortex state is denoted $|\mathbf{p}, l_e\rangle$ and a plane wave by $|\tilde{\mathbf{p}}\rangle$. Note, for the two-dimensional vectors we use $\mathbf{p} = (p_\parallel, p_\perp)$ and for three-dimensional vectors we include a tilde, $\tilde{\mathbf{p}} = (p_\parallel, p_\perp, \phi)$, written in cylindrical coordinates. Here, $p_\perp = \sqrt{p_x^2 + p_y^2}$ is the radial coordinate, $p_\parallel = p_z$ is the momentum coordinate along the cylindrical axis (parallel to the laser field polarization) and $\phi$ the azimuthal angle $\phi = \arctan(p_y/p_x)$, while $l_e$ is the topological charge or azimuthal OAM. We will employ atomic units throughout, unless otherwise stated. The spatial representation of the vortex state is given by[52]

$$\langle \tilde{\mathbf{r}} | \mathbf{p}, l_e \rangle = \frac{1}{(2\pi)^{3/2}} J_{l_e}(p_\perp r_\perp) e^{i\phi l_e} e^{i p_\parallel r_\parallel}, \tag{1}$$

while the momentum representation is

$$\langle \tilde{\mathbf{k}} | \mathbf{p}, l_e \rangle = \frac{i^{-l_e} e^{i\phi' l_e}}{2\pi p_\perp} \delta(k_\parallel - p_\parallel) \delta(k_\perp - p_\perp). \tag{2}$$

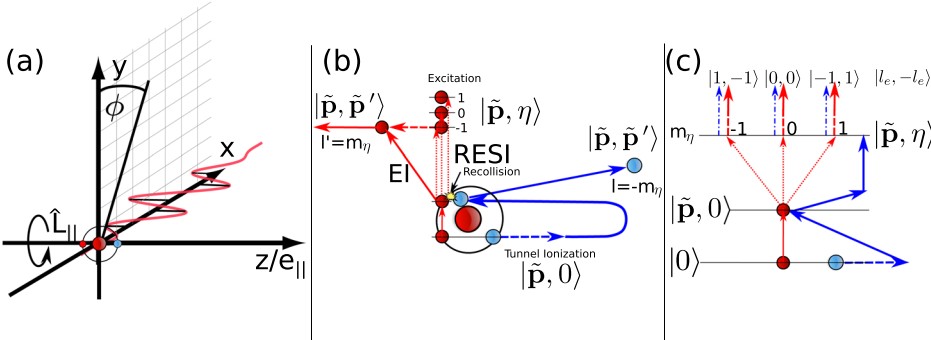

**Fig. 1 | Diagram of NSDI and the resulting entanglement of OAM. a** Orientation of the linearly polarized laser, with polarization $e_\parallel$ along the $z$ axis taken for the OAM denoted by $\hat{L}_\parallel$ and the target atom. **b** NSDI process depicted for the EI and RESI mechanisms. Interaction via the field (e.g. tunnel ionization) is depicted by a dashed line, excitation in the singly charged ion is denoted by dotted lines, while the recollision and OAM sharing is denoted by the yellow spark. The two electron

states are defined in the text, following the convention of ref. [66]. **c** The excitation pathways in RESI, which lead to different final OAM states and an entangled superposition. The final states are given by OAM states, $|l_e, -l_e\rangle$ as used in (8). The dashed-dotted lines are to denote, which two-electron state the first electron ends up in. The numbers −1, 0, 1 refer to the values of the quantum magnetic number $m_\eta$ of the intermediate excited state, $|\tilde{\mathbf{p}}, \eta\rangle$.

We assume in the asymptotic limit of large distances from the atomic nucleus that the final state of the two electrons in the continuum can be written as a product of vortex states

$$|\mathbf{p}, l_e, \mathbf{p}', l_e'\rangle = |\mathbf{p}, l_e\rangle \otimes |\mathbf{p}', l_e'\rangle. \tag{3}$$

Typically for NSDI we consider momentum measurement and the associated transition amplitude

$$M(\tilde{\mathbf{p}}, \tilde{\mathbf{p}}') = \lim_{t \to \infty} \langle \tilde{\mathbf{p}}, \tilde{\mathbf{p}}'|\psi(t)\rangle, \tag{4}$$

where $|\psi(t)\rangle$ denotes the continuum two-electron wavepacket after the interaction with the external field, and $|\tilde{\mathbf{p}}, \tilde{\mathbf{p}}'\rangle$ denotes the scattering state with asymptotic momenta $\tilde{\mathbf{p}}$ and $\tilde{\mathbf{p}}'$. Here, however, we consider the transition amplitude corresponding to OAM measurement, including only the doubly ionized portion of the system, which may be expressed as

$$\begin{aligned}M_{l_e, l_e'}(\mathbf{p}, \mathbf{p}') &= \lim_{t \to \infty} \langle \mathbf{p}, l_e, \mathbf{p}', l_e'|\psi(t)\rangle \\ &= \frac{i^{l_e + l_e'}}{(2\pi)^2} \iint d\phi d\phi' e^{-i\phi l_e - i\phi' l_e'} M(\tilde{\mathbf{p}}, \tilde{\mathbf{p}}'), \end{aligned} \tag{5}$$

where $M_{l_e, l_e'}(\mathbf{p}, \mathbf{p}')$ is also the two-dimensional Fourier series coefficient of $M(\tilde{\mathbf{p}}, \tilde{\mathbf{p}}')$. Note, we will always construct $M_{l_e, l_e'}(\mathbf{p}, \mathbf{p}')$ such that electron indistinguishability is accounted for.

The total azimuthal OAM of the two electrons will be conserved at all times

$$l_e + l_e' = m + m', \tag{6}$$

where $l_e$ and $l_e'$ are the azimuthal OAM of the final vortex state and $m$ and $m'$ are the initial quantum magnetic numbers of the two-electron ground state $|0\rangle$. For the recolliding electron in NSDI $m \neq 0$ is strongly suppressed, so we consider $m = 0$. Similarly, for the second electron contributions from $m'$ with opposite signs will destructively interfere, thus we take $m' = 0$. The OAM conservation is now trivially $l_e' = -l_e$, see Fig. 1(b) and (c). For the RESI mechanism, the second electron leaves from an excited state $|\eta\rangle$, and thus we have the additional conservation $l_e' = m_\eta$, thus $l_e = -m_\eta$, which again can be seen in Fig. 1(b) and (c). It is also important to consider the role of the ion, if different ionic states were associated with different final OAM states, this would lead to decoherence that could reduce entanglement. However, given the selection rules lead to $m = m' = 0$, the residual ion will only feasibly have one final OAM state. Thus, the ion may be traced out without affecting the electron–electron entanglement.

Due to the recollision there is OAM sharing in NSDI, while, in the RESI mechanism, the OAM is tunable via the excited state. Furthermore, the excited electron may occupy a superposition of states[69–72], which means entanglement can emerge in the OAM degree of freedom. From this point onwards, we only consider the excited state populates $m_\eta = 0, \pm 1$. This reduction on the OAM space captures all necessary physics, while limiting the complexity of measurement/implementation. OAM measurement across considerably larger ranges has been achieved in electron beams, see, e.g., refs. [49,53,55,56]. The final two-electron continuum state corresponding to this scenario can be described by,

$$\begin{aligned}|\psi\rangle = \iint d^2\mathbf{p} d^2\mathbf{p}' \big(&M_{1-1}(\mathbf{p}, \mathbf{p}')|\mathbf{p}, 1, \mathbf{p}', -1\rangle \\ &+ M_{00}(\mathbf{p}, \mathbf{p}')|\mathbf{p}, 0, \mathbf{p}', 0\rangle + M_{-11}(\mathbf{p}, \mathbf{p}')|\mathbf{p}, -1, \mathbf{p}', 1\rangle\big), \end{aligned} \tag{7}$$

where we consider a time after the end of the pulse and have suppressed $t$ in our notation for convenience, and we will follow this convention throughout the remainder of this work. Here, $|\psi\rangle$ is a

maximally entangled qutrit if $M_{1-1}(\mathbf{p}, \mathbf{p}') = M_{00}(\mathbf{p}, \mathbf{p}') = M_{-11}(\mathbf{p}, \mathbf{p}')$. In fact, even for arbitrary $M$'s, it is possible to show, see Methods, that after the momentum coordinates are traced out the resulting mixed state is always entangled via the positive partial transpose (PPT) criterion[73,74]. With the condition $\iint d^2\mathbf{p} d^2\mathbf{p}' M_{k-k}(\mathbf{p}, \mathbf{p}') M_{k'-k'}^*(\mathbf{p}, \mathbf{p}') \neq 0 \ \forall \ k, k' \in [-1, 1]$. This means OAM entanglement can survive integration over all the momentum coordinates, as long as the probability of excitation to states with two or more values of $m_\eta$ is non-zero. Entanglement in the EI mechanism is also possible, however, in general the final OAM $l_e = l_e' = 0$ dominates, keeping the entanglement associated with EI relatively low, see Supplementary Information for details.

### Entanglement measure and witness

In order to quantify and measure entanglement, we consider the density matrix, $\rho = |\psi\rangle\langle\psi|$. To greatly simplify matters we will assume that the experimentalists are only interested in measuring the OAM, thus we will compute the reduced density matrix, tracing over all the continuous momentum components, see (36), leaving an entangled mixed state,

$$\rho_{\text{OAM}} = \sum_{l_e, l_e'} \alpha_{l_e l_e'} |l_e, -l_e\rangle\langle l_e', -l_e'|, \tag{8}$$

with

$$\alpha_{l_e l_e'} = \int d^2\mathbf{p} \int d^2\mathbf{p}' M_{l_e - l_e}(\mathbf{p}, \mathbf{p}') M_{l_e' - l_e'}^*(\mathbf{p}, \mathbf{p}'). \tag{9}$$

Note, for all computations the density matrix will be normalized by its trace.

The logarithmic negativity $(E_{\mathcal{N}})$[61] is a measure of entanglement, that is valid for mixed state systems and exploits the PPT separability criterion[73,74] and is easy to compute, which makes it a good choice for our purposes. $E_{\mathcal{N}}$ is given by

$$E_{\mathcal{N}} = \log_2 \left[ ||\rho_{\text{OAM}}^{T_A}||_1 \right], \tag{10}$$

where the trace norm reads

$$||\rho||_1 = \text{Tr} \left[ \sqrt{\rho^\dagger \rho} \right] \tag{11}$$

and $\rho^{T_A}$ is the partial transpose, i.e., the transpose with respect to one subsystem. $E_{\mathcal{N}}$ may vary between 0 and $\log_2(3) \approx 1.58$ for the qutrit system in (7).

This measure exploits the partial transpose $(\rho^{T_A})$, which can be related to time-reversal of one of the subsystems. As such, $\rho^{T_A}$ may not always correspond to a physical system, which manifests by negative eigenvalues. The partial transpose preserves positive eigenvalues for a separable system, but not for entangled systems. There exist entangled systems, where the partial transpose does not yield negative eigenvalues. These will yield a logarithmic negativity of zero. Correspondingly, $E_{\mathcal{N}} = \log_2(2\mathcal{N} - 1)$, where $\mathcal{N}$ (the negativity) is given by the absolute value of the sum of the negative eigenvalues of $\rho^{T_A}$. $E_{\mathcal{N}}$ gives an upper bound to the distillable entanglement, i.e., quantifying the number of copies of $\rho$ required to transform it into a maximally entangled state. Other commonly used entanglement measures, such as entropy of entanglement and the purity, exploit that the reduced density matrix (tracing over one of the subsystems) cannot easily be used if the input state is mixed, for more information, see the Supplementary Information.

To measure if there is entanglement, one approach is to use an entanglement witness, which can distinguish a subset of entangled states as non-separable. They can commonly be associated with an observable for which the expectation value is negative for some

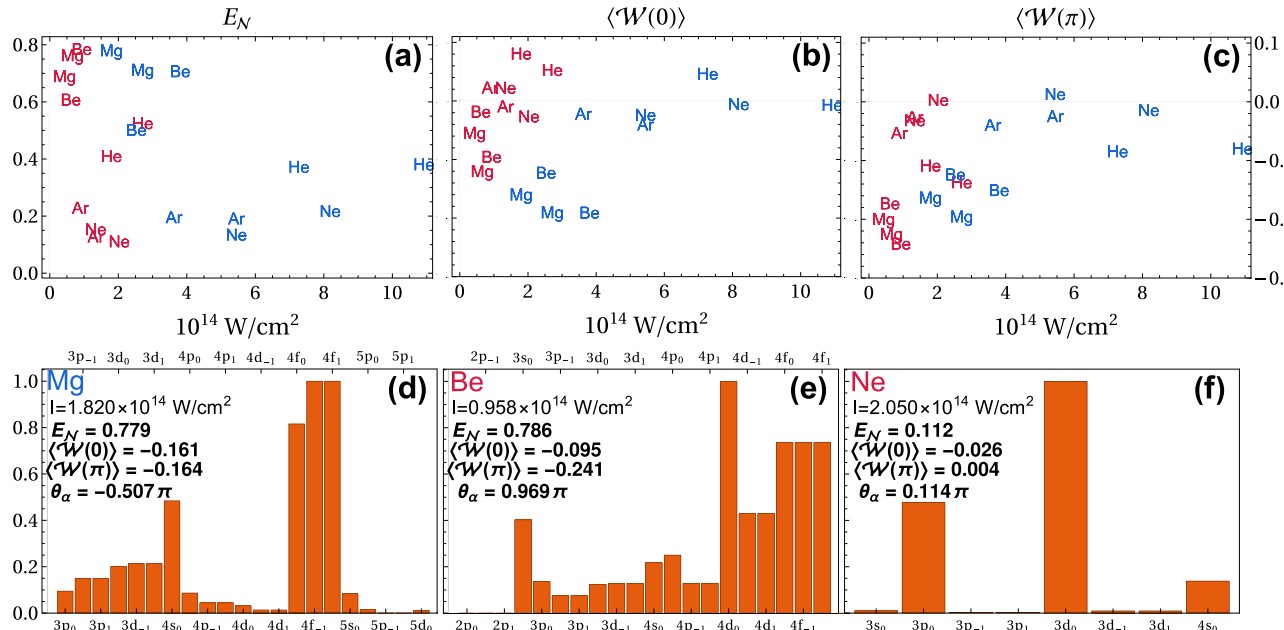

**Fig. 2 | Extensive search over targets and parameters. a** Quantifies the entanglement using the logarithmic negativity ($E_\mathcal{N}$ of (10)). **b** and **c** Expectation value of the entanglement witnesses ($\langle\mathcal{W}(0)\rangle$ and $\langle\mathcal{W}(\pi)\rangle$, see (12)), respectively. Here, $\langle\mathcal{W}\rangle := \mathrm{Tr}[\mathcal{W}\rho_{\mathrm{OAM}}]$. Blue text indicates $\lambda = 400$ nm and red $\lambda = 800$ nm. **d**, **e**, **f** Estimated contribution of intermediate excited states for three extremal cases Mg, Be and Ne, respectively. The field values are given on the panels. The contribution is computed by taking the peak value of the momentum dependent probability of each channel individually. The values are normalized by the largest channel and in arbitrary units. We show $E_\mathcal{N}$, $\langle\mathcal{W}(0)\rangle$, $\langle\mathcal{W}(\pi)\rangle$ and the phase $\theta_\alpha = \arg(\alpha_{10})$, see (8). The excited state labels, at the top and bottom of the panels, are in the format $|\eta\rangle = |n\ell_{m_\eta}\rangle$, where $n$ is the principle quantum number, $\ell$ is the angular quantum number and $m_\eta$ is the magnetic quantum number.

entangled states, i.e., for a witness $\mathcal{W}$ and state $\rho$, the condition $\mathrm{Tr}[\mathcal{W}\rho] < 0$ implies $\rho$ is entangled. It can be shown that

$$\mathcal{W}(\theta) = \frac{1}{d}\mathbb{1} - |\nu(\theta)\rangle\langle\nu(\theta)| \quad (12)$$

with

$$|\nu(\theta)\rangle = \frac{1}{\sqrt{d}}\sum_{l_e=-1}^{1} e^{i\theta l_e}|l_e, -l_e\rangle, \quad (13)$$

is a valid entanglement witness. Here, the dimension $d = 3$. This witness is useful as the state $|\nu(\theta)\rangle$ mirrors the anti-correlated ($l_e' = -l_e$) entangled superposition of the OAM and also contains a system-dependent tuning parameter $\theta$, which can be set depending on the target and field parameters to enhance the detectability. The parameter $\theta$ can be tuned depending on the phase information in the state, in particular depending on $\theta_\alpha := \arg(\alpha_{01})$, where $\alpha_{01}$ represents the coherence between $|0,0\rangle$ and $|\pm1,\mp1\rangle$ in the density matrix $\rho_{\mathrm{OAM}}$ given by (8). We will set $\theta$ to two extremes $\theta = 0$ or $\pi$ and discuss the correspondence with $\theta_\alpha$.

**Optimizing entanglement**

In Fig. 2 we show the logarithmic negativity and the expectation value of the entanglement witness for many targets and laser parameters. We do this using the strong-field approximation (SFA), the details of which are described in the Methods section and refs. 70,71. The SFA is an approximate method, but it captures the basic OAM correlation and is rapid to compute, so is well-suited to a broad parameter search. The parameters are chosen such that the return energy of the first electron, $\approx 3.17 U_p$, is not above the ionization potential of the second, to ensure that the RESI mechanism is dominant. Here, $U_p$ is the ponderomotive or quiver energy of the electron in the laser field. Also, we stay approximately in the tunneling regime, with the Kelydsh[75] parameter $\gamma = \sqrt{I_p/(2U_p)}$ spanning 0.87–1.20. Note that the entanglement is not

dependent on tunnel ionization, i.e., there will still be entanglement if the second electron is ionized through multiphoton absorption.

Immediately from Fig. 2a, it is clear that magnesium and beryllium have the highest logarithmic negativity, while the nobel gases argon and neon have the lowest. We may write the logarithmic negativity in terms of the coefficients $\alpha_{l_e, l_e'}$, $E_\mathcal{N} = \log_2(\alpha_{00} + 4|\alpha_{10}| + 4\alpha_{11})$, due to indistinguishability of the electrons, which means $\alpha_{ij} = \alpha_{-ij} = \alpha_{i-j} = \alpha_{-i-j}$. The coefficients $\alpha_{00}$ and $\alpha_{11}$ are determined by the population of intermediate states with $m_\eta = 0$ and $m_\eta = \pm1$, respectively, while the term $\alpha_{10}$ is a measure of the coherence between these states. In the bottom row, panels (d)–(f), the contribution of channels via different excited states is estimated for the two highest and lowest cases. In the case of the nobel gases, states with $m_\eta = 0$ dominate, thus reducing the entanglement as a single OAM prevails, see panel (f). This selectivity provides information on recollision dynamics: For nobel gases, the second electron initial $p$-state (with $m' = 0$) is aligned along the OAM axis and transitions to excited states which keep this alignment are more probable. For magnesium and beryllium in panels (d) and (e), there is a more balanced superposition across $m_\eta = -1, 0, 1$, the initial $s$-state of these targets is spherically symmetric, so there will be no directional preference for the excited state, leading to higher logarithmic negativity and a higher degree of entanglement. Thus, the logarithmic negativity is a direct probe of excited state population and geometry.

In the panels (b) and (c), the expectation values of $\mathcal{W}(0)$ and $\mathcal{W}(\pi)$ are shown. Values below and above zero correspond to entangled states that may and may not be distinguished from separable states, respectively. The witness $\mathcal{W}(\pi)$ outperforms $\mathcal{W}(0)$, which can be traced back to the phase $\theta_\alpha$, which varies with the target and field parameters. The expectation value of the witnesses can be expressed in terms of the coefficients $\alpha_{ij}$, $\langle\mathcal{W}(0/\pi)\rangle = -\frac{2}{3}(\alpha_{11} \pm 2|\alpha_{10}|\cos(\theta_\alpha))$. Thus, the witness probes the coherence term, $\alpha_{10}$ and its phase $\theta_\alpha$. The witness $\mathcal{W}(\pi)$ improves the detectability of entanglement for $\lambda = 800$ nm, while less difference is seen for $\lambda = 400$ nm. E.g. beryllium in panel (e) $\theta_\alpha \approx \pi$ so the witness $\mathcal{W}(\pi)$ is much lower than $\mathcal{W}(0)$.

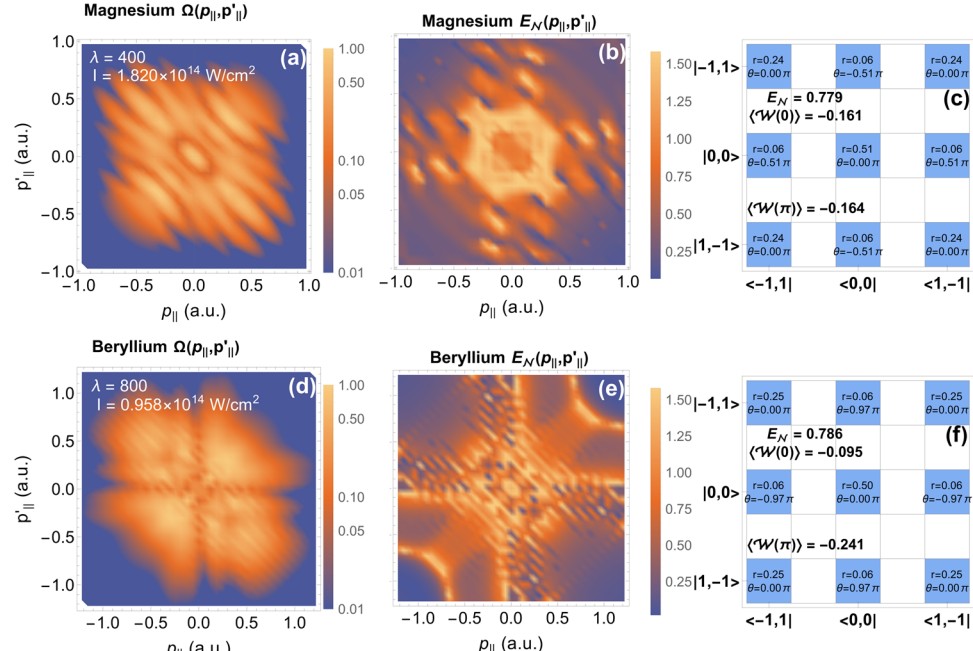

**Fig. 3 | Momentum dependent probability, logarithmic negativity, and density matrices. a** and **d** The correlated momentum distribution $\Omega(p_{\parallel}, p'_{\parallel})$ for magnesium and beryllium, respectively, with respect to the parallel momentum coordinates $p_{\parallel}$ and $p'_{\parallel}$. We have integrated over the components perpendicular to the laser field polarization, given by (14). A logarithmic scale is given, normalized to the peak value in arbitrary units. The target is given in the title, while the field parameters are listed in the top left. **b** and **e** Momentum-dependent logarithmic negativity

$E_{\mathcal{N}}(p_{\parallel}, p'_{\parallel})$ for the same respective targets. The perpendicular components are taken to be (nearly) zero. **c** and **f** Display the density matrices, $\rho_{OAM}$ (8), for the indicated targets. The non-zero complex element of $\rho_{OAM}$, are colored blue and represented by the phase $\theta = \arg(\alpha_{l_e l'_e})$ (recall that $\theta_{\alpha} = \arg(\alpha_{01})$) and the modulus $r = |\alpha_{l_e l'_e}|$. The bra and ket labels, **c** and **f**, correspond to the states $|l_e, l'_e\rangle$, where the $l_e$ and $l'_e$ is the OAM of the two photoelectrons.

Furthermore, for magnesium in panel (d) at $\lambda = 400$ nm $\theta_{\alpha} \approx 0.5\pi$, which is why $\mathcal{W}(0)$ and $\mathcal{W}(\pi)$ give nearly identical results. Thus, the witness also provides key information on coherence between $|0, 0\rangle$ and $|\pm 1, \mp 1\rangle$ in $\rho_{OAM}$ via $\alpha_{10}$, which is not usually accessible given that $\alpha_{10}$ is not an observable. Thus, tuning the witness via the parameter $\theta$, enables efficient entanglement detection for specific targets and fields, as well allows the phases between channels of excitation to be determined.

In Fig. 3, we plot the correlated momentum distribution $\Omega(p_{\parallel}, p'_{\parallel})$ over $p_{\parallel}$ and $p'_{\parallel}$, where we have integrated over the components perpendicular to the laser field polarization.

$$\Omega(p_{\parallel}, p'_{\parallel}) \propto \int\int dp_{\perp} dp'_{\perp} p_{\perp} p'_{\perp} |M(\tilde{\mathbf{p}}, \tilde{\mathbf{p}}')|^2. \quad (14)$$

Interferences can be seen, which is a hallmark of the superposition of excited states[69–72,76]. In the second column, we plot momentum-dependent logarithmic negativity $E_{\mathcal{N}}(p_{\parallel}, p'_{\parallel})$. Here, the logarithmic negativity is computed at specific momentum values, see caption, quantifying the entanglement between the photoelectrons given a specific final momentum.

We may write, $E_{\mathcal{N}}(p_{\parallel}, p'_{\parallel}) = \log_2(|M_{-11} + M_{00} + M_{1-1}|^2)$, where we have used $M_{ij} \equiv M_{ij}(\mathbf{p}, \mathbf{p}')|_{p_{\perp} = p'_{\perp} = \delta p}$, see (5). This coherent mix of OAM channels reveals the importance of their relative phases. Such phases are determined by an interplay between recollision dynamics and ionization from the excited state. As such, $E_{\mathcal{N}}(p_{\parallel}, p'_{\parallel})$ varies considerably, forming entanglement fringes, which reveal the phase between the OAM channels that follow the fringes in the momentum distribution. For $E_{\mathcal{N}}(p_{\parallel}, p'_{\parallel})$ of magnesium, panel (b), there are entanglement maxima either side of the $p_{\parallel}$ and $p'_{\parallel}$ axes, with diagonal fringes and a central ring of high logarithmic negativity. This behavior can be explained through the excitation channels $4s_0$, $4f_0$ and $4f_{\pm 1}$, which contribute the most, see Fig. 2. In Fig. 3 there are peaks in the

entanglement when there is a balanced superposition of channels with $m_{\eta} = -1$, 0 and 1. This occurs where $4f_0$ and $4s_0$ are maximum, which is adjacent to the axis and in the center, respectively. For $E_{\mathcal{N}}(p_{\parallel}, p'_{\parallel})$ of beryllium, panel (e), has the same entanglement maxima either side of the $p_{\parallel}$ and $p'_{\parallel}$ axes as magnesium. This arises from the states $4d_0$, $4d_{\pm 1}$, $4f_0$ and $4f_{\pm 1}$, which leads to the same effect due to on-axis nodes of $4d_0$ and $4f_0$. In this case, the mixing of dominant channels with different $l$ leads to the large peaks on the diagonal (around $p_{\parallel}, p'_{\parallel} \approx \pm 0.8$ a.u.), where the photoelectrons have large correlated momentum.

Through the interplay of excitation channels, we have identified three cases where the two photoelectrons approach the maximum logarithmic negativity. (i) Combination of three channels with equal $\ell$ and $m_{\eta} = \pm 1$, 0, which results in off-axis maxima, i.e., fast and slow entangled photoelectrons. (ii) The combination of an $s$ state with higher $\ell$ channel with $m_{\eta} = \pm 1$, leads to central maxima or two slow entangled photoelectrons. (iii) The combination of two sets of differing $\ell$ channels each with $m_{\eta} = \pm 1$, 0, this leads to two fast pairs of entangled photoelectrons, with a correlated direction. The different types of pairs of highly entangled photoelectrons could be useful as imaging probes accessing different momentum regions, or alternatively as a source of OAM entangled electrons, which can be optimized by tuning the interplay between different channels of excitation, for instance with tailored fields.

In Fig. 3, we display $\rho_{OAM}$ for beryllium and magnesium targets. The phases of the complex entries of $\rho_{OAM}$ are written on each element, it is clearly only $\theta_{\alpha}$, the phase between $|\pm 1, \mp 1\rangle$ and $|0, 0\rangle$, that plays a role. The closer the tuning parameter $\theta$ is to $\theta_{\alpha}$ the more effective the entanglement witnesses. The measure $E_{\mathcal{N}}$, on the other hand, is independent of this phase and is determined by the relative magnitude of the non-zero elements of $\rho_{OAM}$. We can see in the figure, it is the elements related to $|\pm 1, \mp 1\rangle$ and $|0, 0\rangle$ that are most reduced. The coherence between these states is reduced after tracing over the momentum coordinates. The coherence between $|\pm 1, \mp 1\rangle$ and $|\mp 1, \pm 1\rangle$

is robust, however, as the channels that leads to these states such as $4d_{\pm 1}$ are degenerate, so will have the same behavior over momentum. Thus, the final mixed state keeps a reasonably high logarithmic negativity.

## Measurement considerations

There are additional incoherent averaging effects leading to mixed states, which will primarily be in the form of fluctuations and averaging over the carrier envelope phase and the laser intensity. For simplicity, we are assuming long enough pulses for the former to not be an issue. The latter takes place through intensity variation over focal volume (focal averaging) as well as intensity variation from shot-to-shot. In the Supplementary Information, we compute the focal averaged momentum distributions and density matrices, following the procedure set out in refs. [71,77]. We find that this does not have a significant effect on the overall entanglement, as the coherence between the states relating to $m_\eta = \pm 1$ is robust to intensity variation.

Aside from incoherent averaging, ion–photoelectron entanglement would lead to decoherence. However, the strict selection rules, prevent decoherence by ensuring there is only one final state of the ion. One could speculate whether a multielectron treatment is required, beyond the 2e + ion model employed here? The remaining core electron in the cases of beryllium and magnesium are tightly bound and unlikely to play a role. In argon these effects may have to be considered. It is possible this would provide additional decoherence channels to final OAM states, which would reduce the entanglement. Full inclusion of the Coulomb potential is also an important consideration. Recent work demonstrated the inclusion of the Coulomb potential introduced recolliding trajectories for the second electron[78]. However, this treatment uses the same S-matrix treatment for the recollision-excitation step, allowing for entanglement via the same mechanism. The ion could couple to the photoelectrons via its angular momenta. Given the large mass of the ion, this coupling is expected to have a vanishing effect.

In its current form, (12), the expectation value of the witness cannot be easily measured as it would require a combined measurement on both particles. However, any witness of this type may be decomposed[79] into a series of local measurements performed on each particle separately[62,63]. Practically, this will mean doing multiple experiments to compute expectation values and then combining these results with suitable weights determined by the decomposition. For the specific values of $\theta = 0$ and $\theta = \pi$, the witness decomposition is

$$\mathcal{W}(0/\pi) = \left[ \frac{8}{3}\mathbb{1}^{\otimes 2} - \lambda_3^{\otimes 2} - 2(\lambda_4^{\otimes 2} + \lambda_5^{\otimes 2}) + \lambda_8^{\otimes 2} + \sqrt{3}(\lambda_3 \otimes \lambda_8 + \lambda_8 \otimes \lambda_3) \right.$$
$$\left. \pm 2(\lambda_1 \otimes \lambda_6 + \lambda_6 \otimes \lambda_1 + \lambda_2 \otimes \lambda_7 + \lambda_7 \otimes \lambda_2) \right]/12, \tag{15}$$

where − is taken for $\theta = 0$ and + is taken for $\theta = \pi$. Here, $\mathbb{1}$ corresponds to the identity, i.e., performing no measurement, while $\lambda_i$ are matrices that label 8 different single-particle measurements. In the case of a two-level system (qubit/ SU(2)) the local measurement may correspond to the Pauli matrices plus the identity, which have a simple geometrical interpretation. For our three-level system (qutrit/ SU(3)) we have used the eight Gell-Mann matrices. As an example, $\lambda_3 = L_\parallel^{(1)}$ corresponds to a direct OAM measurement, while $\lambda_4$ corresponds to measuring in the basis $(|-1\rangle \pm |1\rangle)$ and $|0\rangle$. For a full definition of the Gell–Mann matrices, see the Methods section or ref. [80].

We can use the decomposition (15) to determine how it could reduce the total number of measurements required. There are 11 combinations of $\lambda_x \otimes \lambda_y$. However, we can discount $\mathbb{1}^{\otimes 2}$ as this corresponds to doing nothing. Furthermore, $[\lambda_3, \lambda_8] = 0$ (i.e., they share eigenstates) so all parts containing only $\lambda_3$ and $\lambda_8$ can be determined with the same measurement. Additionally, measurements of the form $\lambda_x \otimes \lambda_y$ and $\lambda_y \otimes \lambda_x$ may be collected simultaneously, as we do not distinguish the two electrons. This leaves five measurement settings, a large reduction to the 81 combinations of measurement for a general qutrit system.

## Entanglement-enhanced attosecond imaging

In the non-perturbative regime, we have shown entanglement can be generated from a recollision process, which protects the electrons from decoherence with the ion. Now we will examine the possibility of exploiting this entanglement for enhanced attosecond imaging, to access information beyond that available in a classically correlated system. Quantum enhancement in attoscience is relatively unexplored, but coherence has been shown to improve measurement precision[81]. To exploit the entanglement, we may measure each electron in a different basis, e.g., OAM (vortex state) and momentum (plane wave) measurements. The probability of this mixed measurement is $\text{Prob}(\mathbf{p}, l, \tilde{\mathbf{p}}') = |M_{l-l}(\mathbf{p}, \tilde{\mathbf{p}}')|^2$, where due to the OAM correlation, we may disentangle channels of excitation. This is in contrast to the purely momentum-based measurement, which leads to $\text{Prob}(\tilde{\mathbf{p}}, \tilde{\mathbf{p}}') = |\sum_{l=-1}^{1} M_{l-l}(\mathbf{p}, \mathbf{p}')|^2$; note for a classically correlated system, this would be an incoherent sum. We may combine such measurements, e.g.,

$$\chi(\tilde{\mathbf{p}}, \tilde{\mathbf{p}}') \equiv \text{Prob}(\tilde{\mathbf{p}}, \tilde{\mathbf{p}}') - \sum_l \text{Prob}(\mathbf{p}, l, \tilde{\mathbf{p}}'). \tag{16}$$

In a separable classically correlated system $\chi(\tilde{\mathbf{p}}, \tilde{\mathbf{p}}') = 0$, while in our entangled system we have $\chi(\tilde{\mathbf{p}}, \tilde{\mathbf{p}}') = \sum_{l \neq l'} M_{l-l}(\mathbf{p}, \mathbf{p}') M_{l'-l'}^*(\mathbf{p}, \mathbf{p}')$, giving access to the correlation terms. Mixed measurements in other bases, such as those defined by the Gell–Mann matrices, would provide yet further correlated measurements. Switching between a coherent and incoherent sum bears similarity to entangled double slit interference observed in doubly ionized $H_2$[82]. By measuring the OAM, we are able to (i) disentangle the channels of excitation with differing quantum magnetic numbers, not possible in an uncorrelated system, and (ii) access coherence terms, yielding phase information not accessible in a classically correlated system. Results, which include the Coulomb potential in the electron propagation for RESI[78], show that recollision of the second electron plays an important role. Thus, trajectories will interfere with differing amounts of interaction with the Coulomb potential, providing holographic interferences[15]. The combination of correlated mixed measurement, exploiting OAM entanglement and photoelectron holography provides two ways to extract phase information, making a powerful attosecond imaging tool.

## Discussion

In this work we use the correlated process of non-sequential double ionization (NSDI), as a backbone, to generate entanglement in the orbital angular momentum (OAM) of two photoelectrons. The photoelectron OAM in NSDI has some general rules, which show that entanglement occurs in the recollision with subsequent ionization (RESI) mechanism of NSDI, prevalent at lower intensities[60]. The OAM in NSDI has not been explored before, and previous entanglement studies have focused on the momentum coordinates parallel to the laser field[33]. We find that OAM entanglement can be maximal in specific momentum regions, and that this is controlled by the interplay of channels of excitation, which may be investigated as a source of OAM-entangled electrons. Furthermore, the entanglement is robust as it survives incoherent averaging over the focal volume and it is not generated by the symmetrization resulting from the indistinguishability of the photoelectrons[83].

The use of the OAM has many benefits, firstly, the entanglement is simply understood as a consequence of angular momentum sharing during recollision, coupled with a superposition over OAM states due to contribution from excitation-channels with a differing quantum magnetic number [Fig. 1c]. Secondly, the quantization enables a clear and simple analysis using the logarithmic negativity, which enables an extensive search over targets and parameters to maximize the

entanglement, where it is clear ideal targets are those with two *s*-state valence electrons, given that this will promote a balanced superposition across OAM states [Fig. 2]. Thirdly, the reduction in computational difficulty allows the density matrix and momentum-dependent logarithmic negativity to be computed [Fig. 3] and enables incoherent averaging. Finally, we can construct an entanglement witness, which may be decomposed into local measurements, avoiding full state tomography or the measurements of incompatible continuous observables, like position and momentum, reducing the difficulty of experimental implementation for the detection of entanglement.

A key question is how to perform such an experiment? The measurement of OAM for electron vortex beams, has received a great deal of attention and a variety of methods have been demonstrated[49–56]. This includes diffractive methods such as a fork hologram[49,50] or Dammann vortex grating[55] or conformal mapping techniques using phase plates[53] or electrostatic fields[56]. This arsenal of techniques is capable of detecting an OAM range far in excess of what is required here. The latter electrostatic OAM sorter[56] boasts high efficiency and was retrofitted as an OAM analyzing element to a transmission electron microscope. Thus, such an element could be conceivably added to a typical reaction microscope (ReMi)[84,85] employed for NSDI—a ReMi measures correlation of the ion and the two electrons, thus, it is already well-suited for studying entanglement[86]. Beyond the measurement of OAM, there will be some difficulty in using alkaline earth metals in experiment over the typical nobel gas targets. Works[87,88] have employed a range of alkaline earth metals in studies on ion rates, even at a longer wavelength of 2000 nm[88], however, a ReMi-style measurement of the electrons' momentum would be challenging. An alternative route could be to investigate larger parameter ranges and tailored fields in order to boost the entanglement in the noble gases or other simpler targets. This search would be aided if the theoretical model included the electron impact mechanism to accurately address the effect on overall entanglement.

The entangled OAM of the photoelectrons in NSDI may be exploited in various ways, e.g., through interferometric schemes exploiting tailored fields, as a source of OAM-entangled electron pairs, or correlated measurements of OAM and momenta. An interesting consideration is the spin of the electrons. Previous work observed spin polarization from the ground states of heavier targets, see e.g.,[89,90]. However, for the targets we have employed this effect will be small. The spin–orbit coupling during strong-field dynamics, e.g., during recollision, may play some role but has been generally neglected in strong-field studies. OAM in attosecond processes provides a rich burgeoning research area, to help achieve the aim of imaging and controlling matter on ultrafast times scales and photoelectron OAM-entanglement plays an important role in achieving this goal. Beyond this, like ref. [20], it further demonstrates the fundamental non-classicality of such processes.

## Methods
### Strong-field approximation
The wave function for NSDI, which describes both electron-impact ionization (EI) and recollision with subsequent ionization (RESI) can be written using an SFA flavor ansatz[66] in the following way

$$
\begin{aligned}
|\psi(t)\rangle = a(t)|0\rangle + \int d^3\tilde{\mathbf{p}}\, b(\tilde{\mathbf{p}},t)|\tilde{\mathbf{p}},0\rangle + \sum_{\eta} \int d^3\tilde{\mathbf{p}}\, c(\tilde{\mathbf{p}},\eta,t)|\tilde{\mathbf{p}},\eta\rangle \\
+ \iint d^3\tilde{\mathbf{p}}\, d^3\tilde{\mathbf{p}}'\, d(\tilde{\mathbf{p}},\tilde{\mathbf{p}}',t)|\tilde{\mathbf{p}},\tilde{\mathbf{p}}'\rangle,
\end{aligned}
\tag{17}
$$

where $|0\rangle$ is the two electron ground state, $|\tilde{\mathbf{p}},0\rangle$ corresponds to one electron in the continuum and the other in its ground state, $|\tilde{\mathbf{p}},\eta\rangle$ is the same as the latter with the second electron excited and $|\tilde{\mathbf{p}},\tilde{\mathbf{p}}'\rangle$ is the two electron continuum state.

The final symmetrized transition amplitude is related to (17) via

$$
\begin{aligned}
M^{\text{RESI}}(\tilde{\mathbf{p}},\tilde{\mathbf{p}}') &= \lim_{t\to\infty} d(\tilde{\mathbf{p}},\tilde{\mathbf{p}}',t) \\
&= \frac{1}{\sqrt{2}}\left( M^{\text{RESI}}_{\text{unsym}}(\tilde{\mathbf{p}},\tilde{\mathbf{p}}') + M^{\text{RESI}}_{\text{unsym}}(\tilde{\mathbf{p}}',\tilde{\mathbf{p}}) \right),
\end{aligned}
\tag{18}
$$

where $M_{\text{unsym}}(\tilde{\mathbf{p}},\tilde{\mathbf{p}}')$ is the unsymmetrized transition amplitude. This is valid for an initial singlet state, as considered here. In the SFA, in atomic units, for the RESI mechanism and using the assumptions listed in refs. [70,71], the unsymmetrized transition amplitude can be written as

$$
M^{\text{RESI}}_{\text{unsym}}(\tilde{\mathbf{p}},\tilde{\mathbf{p}}') = \sum_{\eta} \int d^3 t \int d^3\tilde{\mathbf{k}}\, V_{\tilde{\mathbf{p}}\tilde{\mathbf{p}}',\tilde{\mathbf{p}}\eta} V_{\tilde{\mathbf{p}}\eta,\tilde{\mathbf{k}}0} V_{\tilde{\mathbf{k}}0,0}\, \exp[iS(\mathbf{p},\mathbf{p}',\mathbf{k},t,t',t'')],
\tag{19}
$$

where

$$
\int d^3 t \equiv \int_{-\infty}^{\infty} dt \int_{-\infty}^{t} dt' \int_{\infty}^{t'} dt''
\tag{20}
$$

and

$$
\begin{aligned}
S(&\mathbf{p},\mathbf{p}',\mathbf{k},t,t',t'') \\
&= I_{\text{p}}^{10} t'' + I_{\text{p}}^{20} t' + I_{\text{p}}^{\eta}(t-t') - \int_{t''}^{t'} \frac{[\mathbf{k}+\mathbf{A}(\tau)]^2}{2} d\tau \\
&\quad - \int_{t'}^{\infty} \frac{[\mathbf{p}+\mathbf{A}(\tau)]^2}{2} d\tau - \int_{t}^{\infty} \frac{[\mathbf{p}'+\mathbf{A}(\tau)]^2}{2} d\tau
\end{aligned}
\tag{21}
$$

denotes the semiclassical action and $I_{\text{p}}^{10}$, $I_{\text{p}}^{20}$ and $I_{\text{p}}^{\eta}$ are the one-electron ionization potentials corresponding to removing a bound electron from $|0\rangle$, $|\tilde{\mathbf{p}},0\rangle$ and $|\tilde{\mathbf{p}},\eta\rangle$, respectively. Note that for three-dimensional vectors we include a tilde $\tilde{\mathbf{p}}=(p_{\parallel},p_{\perp},\phi)$, while for two-dimensional vectors we do not $\mathbf{p}=(p_{\parallel},p_{\perp})$, where $p_{\parallel}=p_z$, $p_{\perp}=\sqrt{p_x^2+p_y^2}$ and $\phi$ is the azimuthal angle. The prefactors are given by

$$
\begin{aligned}
V_{\tilde{\mathbf{k}}0,0} &= \langle \tilde{\mathbf{k}}(t''),0|V|0\rangle \\
&= \frac{1}{(2\pi)^{3/2}} \int d^3\tilde{\mathbf{r}}\, V(\tilde{\mathbf{r}}) e^{-i\tilde{\mathbf{k}}(t'')\cdot\tilde{\mathbf{r}}} \psi_{10}(\tilde{\mathbf{r}}),
\end{aligned}
\tag{22}
$$

$$
\begin{aligned}
V_{\tilde{\mathbf{p}}\eta,\tilde{\mathbf{k}}0} &= \langle \tilde{\mathbf{p}}(t'),\eta|V_{12}|\tilde{\mathbf{k}}(t'),0\rangle \\
&= \frac{1}{(2\pi)^3} \iint d^3\tilde{\mathbf{r}}'\, d^3\tilde{\mathbf{r}}\, \exp[-i(\tilde{\mathbf{p}}-\tilde{\mathbf{k}})\cdot\tilde{\mathbf{r}}] \\
&\quad \times V_{12}(\tilde{\mathbf{r}},\tilde{\mathbf{r}}')[\psi_{\eta}(\tilde{\mathbf{r}}')]^* \psi_{20}(\tilde{\mathbf{r}}')
\end{aligned}
\tag{23}
$$

and

$$
\begin{aligned}
V_{\tilde{\mathbf{p}}\tilde{\mathbf{p}}',\tilde{\mathbf{p}}\eta} &= \langle \tilde{\mathbf{p}}(t),\tilde{\mathbf{p}}'(t)|V_{\text{ion}}|\tilde{\mathbf{p}}(t),\eta\rangle \\
&= \frac{1}{(2\pi)^{3/2}} \int d^3\tilde{\mathbf{r}}'\, V_{\text{ion}}(\tilde{\mathbf{r}}') e^{-i\tilde{\mathbf{p}}'(t)\cdot\tilde{\mathbf{r}}'} \psi_{\eta}(\tilde{\mathbf{r}}'),
\end{aligned}
\tag{24}
$$

where $\tilde{\mathbf{p}}(t)$, $\tilde{\mathbf{p}}'(t)$ and $\tilde{\mathbf{k}}(t)$ are defined according to $\tilde{\mathbf{k}}(t)=\tilde{\mathbf{k}}+\tilde{\mathbf{A}}(t)$ or $\tilde{\mathbf{k}}(t)=\tilde{\mathbf{k}}$ in the length or velocity gauge, respectively. In this work, for simplicity, we employ the velocity gauge. This formalism describes the RESI process, in which an electron is ionized by the laser field from the ground state $|0\rangle$ at time $t''$ into $|\tilde{\mathbf{k}},0\rangle$ with intermediate momentum $\tilde{\mathbf{k}}$, it recollides at $t'$ and excites a second electron into the state $|\tilde{\mathbf{p}},\eta\rangle$ with a final momentum $\mathbf{p}$ for the initial electron. The second electron is ionized via the laser field at time $t$ into the state $|\tilde{\mathbf{p}},\tilde{\mathbf{p}}'\rangle$ with a final momentum $\tilde{\mathbf{p}}'$. The prefactors give the information about all the bound states[91] and interactions, for which we employ $V$ the singly charged binding potential for the first electron, $V_{\text{ion}}$ the doubly charged binding

potential for the second electron and $V_{12}$ the electron–electron interaction. In this approximation, electron–electron correlation is described by the prefactor $V_{\tilde{\mathbf{p}}\eta,\tilde{\mathbf{k}}0}$. The transition amplitude (19) is computed using the steepest descent method. In which, we look for values of the variables $t, t', t''$ and $\tilde{\mathbf{k}}$ such that the action is stationary. This leads to the following saddle-point equations

$$\left[\tilde{\mathbf{k}} + \tilde{\mathbf{A}}(t'')\right]^2 = -2I_{\mathrm{p}}^{10}, \tag{25}$$

$$\tilde{\mathbf{k}} = -\frac{1}{t' - t''}\int_{t''}^{t'} d\tau\, \tilde{\mathbf{A}}(\tau), \tag{26}$$

$$\left[\tilde{\mathbf{p}} + \tilde{\mathbf{A}}(t')\right]^2 = \left[\tilde{\mathbf{k}} + \tilde{\mathbf{A}}(t')\right]^2 - 2(I_{\mathrm{p}}^{20} - I_{\mathrm{p}}^{\eta}), \tag{27}$$

and

$$\left[\tilde{\mathbf{p}}' + \tilde{\mathbf{A}}(t)\right]^2 = -2I_{\mathrm{p}}^{\eta}. \tag{28}$$

Eqs. (25) and (28) give the energy conservation of the first and second electron at the instant of tunnel ionization, (26) enforced the first electron will return and (27) describe energy sharing between the electrons.

## Conservation laws

The conservation laws exploited in the main text are general and can be arrived at with few assumptions. The Hamiltonian of the two-electron system may be written as

$$\hat{H}(t) = \frac{\hat{p}^2 + \hat{p}'^2}{2} + (r_{\parallel} + r'_{\parallel})E_{\parallel}(t) + V_{\mathrm{ion}}(\tilde{\mathbf{r}}) + V_{\mathrm{ion}}(\tilde{\mathbf{r}}') + \hat{V}_{12}. \tag{29}$$

All terms are independent of the coordinates $\phi$ and $\phi'$ except for $\hat{V}_{12}$, which depends on the relative distance between the electrons $|\tilde{\mathbf{r}} - \tilde{\mathbf{r}}'| = \sqrt{(r_{\parallel} - r'_{\parallel})^2 + r_{\perp}^2 r'^2_{\perp}\cos(\phi - \phi')}$, thus this is still invariant to a rotation to both particles. Hence, $[\hat{L}_{\parallel}, \hat{H}] = 0$, where $\hat{L}_{\parallel} = -i\partial_{\Phi} - i\partial_{\Phi'}$, and total OAM is conserved, $m + m' = l_{\mathrm{e}} + l'_{\mathrm{e}}$. If, during ionization by the laser field, the single-active-electron approximation is employed (as is widespread) orbital angular momentum will be conserved during ionization, given that $[\hat{L}_{\parallel}^{(1)}, \hat{H}^{(1)}] = [\hat{L}_{\parallel}^{(2)}, \hat{H}^{(2)}] = 0$, where $\hat{L}^{(1)}$, $\hat{L}^{(2)}$, $\hat{H}^{(1)}$ and $\hat{H}^{(2)}$ are the OAM operators and Hamiltonians for each individual electron, including only one-particle terms. Thus, this lead to $l'_{\mathrm{e}} = m_{\eta}$.

The same conservation laws plus additional constraints are encoded in the SFA via the prefactors, which may be written in terms of their dependence on the azimuthal angles,

$$V_{\tilde{\mathbf{k}}0,0} = e^{im\phi_{\mathbf{k}}}\tilde{V}_{\mathbf{k}0,0} \tag{30}$$

$$V_{\tilde{\mathbf{p}}\eta,\tilde{\mathbf{k}}0} = e^{i(m'-m_{\eta})\phi_{\mathbf{p}}}\tilde{V}_{\mathbf{p}\eta,\mathbf{p}''0} \tag{31}$$

$$V_{\tilde{\mathbf{p}}\tilde{\mathbf{p}}',\tilde{\mathbf{p}}\eta} = e^{im_{\eta}\phi_{\mathbf{p}'}}\tilde{V}_{\mathbf{p}\mathbf{p}',\tilde{\mathbf{p}}\eta}, \tag{32}$$

where the tilde indicates quantities independent of the azimuthal angle of all coordinates. Now using (5) and the above equations we may write the SFA OAM transition amplitude

$$M_{l_{\mathrm{e}},l'_{\mathrm{e}}}^{\mathrm{RESI}}(\mathbf{p},\mathbf{p}') = i^{-(l_{\mathrm{e}}+l'_{\mathrm{e}})}\delta_{m,0}\delta_{m'-m_{\eta},l_{\mathrm{e}}}\delta_{m_{\eta},l'_{\mathrm{e}}}\tilde{M}(\mathbf{p},\mathbf{p}') \tag{33}$$

with

$$\tilde{M}(\mathbf{p},\mathbf{p}') = \int d^3t \int d^2\mathbf{k}\, \tilde{V}_{\mathbf{p}\mathbf{p}',\mathbf{p}\eta}\tilde{V}_{\mathbf{p}\eta,\mathbf{k}0}\tilde{V}_{\mathbf{k}0,0}\, e^{iS(\mathbf{p},\mathbf{p}',\mathbf{k},t,t',t'')}. \tag{34}$$

Thus, we recover the above stated conservation equations along with the condition $m = 0$. The second electron in its ground states has quantum magnetic number $m'$, and from the behavior of the spherical harmonic $Y_{\ell_{\eta}}^{-m'}(\Omega) = (-1)^{m'}Y_{\ell_{\eta}}^{m'}(\Omega)$ we can deduce that, for odd values of $m'$ we will get opposite signs in the final transition amplitude and thus odd pairs will cancel, as the initial states will be degenerate. Thus, for the $s$ and $p$ initial states employed here, we can assume $m' = 0$.

## Density matrix

The full density matrix $\rho = |\psi\rangle\langle\psi|$, where $|\psi\rangle$ is given by (7), is

$$\rho = \sum_{l_{\mathrm{e}},l'_{\mathrm{e}}}\iint d^2\mathbf{p}\, d^2\mathbf{p}'\, d^2\mathbf{p}''\, d^2\mathbf{p}'''\, M_{l_{\mathrm{e}},-l_{\mathrm{e}}}(\mathbf{p},\mathbf{p}')M^*_{l'_{\mathrm{e}}-l'_{\mathrm{e}}}(\mathbf{p}'',\mathbf{p}''')|\mathbf{p},l_{\mathrm{e}},\mathbf{p}',-l_{\mathrm{e}}\rangle\langle\mathbf{p}'',l'_{\mathrm{e}},\mathbf{p}''',-l'_{\mathrm{e}}|. \tag{35}$$

We do not compute this explicitly due to the continuous momentum coordinates, instead most commonly, we will trace out the momentum coordinates

$$\rho_{\mathrm{OAM}} = \iint d^2\mathbf{k}\, d^2\mathbf{k}'\langle\mathbf{k},\mathbf{k}'|\rho|\mathbf{k},\mathbf{k}'\rangle, \tag{36}$$

where we assume $\langle\mathbf{k}|\mathbf{p},l_{\mathrm{e}}\rangle = \delta(\mathbf{k}-\mathbf{p})|l_{\mathrm{e}}\rangle$. Here $|l_{\mathrm{e}}\rangle$ is an OAM state with the property $\langle r|l_{\mathrm{e}}\rangle \propto e^{il_{\mathrm{e}}\phi}$. Applying this rule results in (8). In order to compute the momentum dependent logarithmic negativity $E_{\mathcal{N}}(p_{\parallel},p'_{\parallel})$ we need to compute the density matrix conditioned on some specific final momentum

$$\rho(\mathbf{p},\mathbf{p}') = \sum_{l_{\mathrm{e}},l'_{\mathrm{e}}} M_{l_{\mathrm{e}},-l_{\mathrm{e}}}(\mathbf{p},\mathbf{p}')M^*_{l'_{\mathrm{e}},-l'_{\mathrm{e}}}(\mathbf{p},\mathbf{p}')|\mathbf{p},l_{\mathrm{e}},\mathbf{p}',-l_{\mathrm{e}}\rangle\langle\mathbf{p},l'_{\mathrm{e}},\mathbf{p}',-l'_{\mathrm{e}}|. \tag{37}$$

## Positive partial transpose

The positive partial transpose (PPT) or Peres-Horodecki criterion[73,74] is a necessary condition on density matrices to determine if a system is separable. It is valid for both pure and mixed states. The approach is to take the partial transpose—i.e., transpose one subsystem—and compute the eigenvalues, if any are negative the state is non-separable and thus, entangled. For $2 \otimes 2$ and $2 \otimes 3$ the condition is also sufficient, so no negative eigenvalues imply separability, for higher dimensional systems (such as our $3 \otimes 3$ system), this is not the case, however, it is still a very powerful method and witnesses can be constructed to detect any PPT entangled state[92], which we can exploit to our advantage.

For our NSDI qutrit mixed state

$$\rho_{\mathrm{OAM}} = \sum_{l_{\mathrm{e}},l'_{\mathrm{e}}=-1}^{1}\alpha_{l_{\mathrm{e}}l'_{\mathrm{e}}}|l_{\mathrm{e}},-l_{\mathrm{e}}\rangle\langle l'_{\mathrm{e}},-l'_{\mathrm{e}}|, \tag{38}$$

where $\alpha$ is defined as in (8). In taking the partial transpose we swap the indices for one of the subsystems (in this case the second electron),

$$\rho_{\mathrm{OAM}} = \sum_{l_{\mathrm{e}},l'_{\mathrm{e}}=-1}^{1}\alpha_{l_{\mathrm{e}}l'_{\mathrm{e}}}|l_{\mathrm{e}},-l'_{\mathrm{e}}\rangle\langle l'_{\mathrm{e}},-l_{\mathrm{e}}|. \tag{39}$$

The eigenvalues can be computed analytically as

$$\alpha_{l_{\mathrm{e}}l_{\mathrm{e}}} = \int d^2\mathbf{p}\int d^2\mathbf{p}'|M_{l_{\mathrm{e}}-l_{\mathrm{e}}}(\mathbf{p},\mathbf{p}')|^2 \text{ for } l_{\mathrm{e}} \in [-1,1], \tag{40}$$

which are positive and $\pm|\alpha_{l_e l'_e}|$ for $l_e \neq l'_e$, which provides three negative eigenvalues as long as $|\alpha_{l_e l'_e}| \neq 0$ for $l_e \neq l'_e$. Thus, the electrons from the RESI mechanism of NSDI are always PPT entangled as long as there is non-zero population across different excited state with differing $m_n$. This also means there will always be an entanglement witness, which may be used to experimentally verify this entanglement.

## Logarithmic negativity

The logarithmic negativity is well-suited for quantifying entanglement in PPT entangled states, as it is equal to the entanglement cost to create this entanglement via PPT operations. In the main article we construct the logarithmic negativity from the reduced density matrix, traced over momentum coordinates, however, we may instead define a momentum dependent logarithmic negativity

$$E_{\mathcal{N}}(p_{\parallel}, p') = \log_2 \left[ ||\rho^{T_A}(p_{\parallel}, \delta p, p', \delta p)||_1 \right], \quad (41)$$

where the perpendicular momentum coordinates are set to nearly to zero, $\delta p = 0.05$ a.u., in order allow for 2D visualization. The value is chosen to be where the distribution will have high probability but no nodes due to the geometry of the excited states.

The logarithmic negativity can be related to the sum of the negative eigenvalues ($\lambda_i$) of the density matrix

$$E_{\mathcal{N}} = \log_2 \left[ ||\rho^{T_A}_{\text{OAM}}||_1 \right]$$
$$= \log_2 \left( 1 + 2 | \sum_{\lambda_i < 0} \lambda_i | \right) \quad (42)$$

which can be written as minus the sum of the negativive eigenvalues

$$= \log_2 \left( 1 + 2 ( |\alpha_{-10}| + |\alpha_{-11}| + |\alpha_{01}| ) \right) \quad (43)$$

Using the Cauchy–Schwartz inequality we can show $\alpha_{l_e l'_e} \leq \sqrt{\alpha_{l_e l_e} \alpha_{l'_e l'_e}}$, leading to

$$
\begin{aligned}
&|\alpha_{-10}| + |\alpha_{-11}| + |\alpha_{01}| \\
&\leq \sqrt{\alpha_{-1-1} \alpha_{00}} + \sqrt{\alpha_{-1-1} \alpha_{11}} + \sqrt{\alpha_{00} \alpha_{11}} \\
&\leq \sqrt{\alpha_{-1-1} + \alpha_{11} + \alpha_{00}} \sqrt{\alpha_{00} + \alpha_{-1-1} + \alpha_{11}} \\
&= 1,
\end{aligned}
\quad (44)
$$

where other forms of the Cauchy–Schwartz inequality were used in the additional inequalities. With this we can place a bound upon the logarithmic negativity

$$E_{\mathcal{N}} \leq \log_2(1+2) \approx 1.58 . \quad (45)$$

## Entanglement witnesses

Here, we will show that the witness used in this work (see (12)) is a valid entanglement witness. We do this by showing that (i) it is positive for all separable states and (ii) the trace is negative for at least one entangled state. For a general separable pure state

$$|\psi\rangle = \sum_i a_i |i\rangle \otimes \sum_j b_j |j\rangle \quad (46)$$

the density matrix may be written as

$$\rho_s = \sum_{i,j,m,n} a_i a^*_m b_j b^*_n |i,j\rangle\langle m,n| \quad (47)$$

and we can compute the trace with the witness of (12)

$$\text{Tr}[\rho_s \mathcal{W}(\theta)] = \frac{1}{d} \sum_{i,j} |a_i|^2 |b_j|^2 - \langle \nu(\theta)|\rho_s|\nu(\theta)\rangle \quad (48)$$

$$= \frac{1}{d} - | \sum_{ij} a_i b_j \langle i,j|\nu(\theta)\rangle |^2 \quad (49)$$

$$= \frac{1}{d} - \frac{1}{d} | \sum_{l_e} a_{l_e} e^{i\theta l} b_{-l_e} |^2 \quad (50)$$

$$\geq \frac{1}{d} - \frac{1}{d} \sum_{l_e} |a_{l_e} e^{i\theta l_e}|^2 \sum_{l_e} |b_{l_e}|^2 = 0. \quad (51)$$

Thus, we have demonstrated (i) and the entangled state we use for (ii) is $|\nu(\theta)\rangle$

$$\text{Tr}[|\nu(\theta)\rangle\langle\nu(\theta)|\mathcal{W}(\theta)] = \frac{1}{d} - |\langle\nu(\theta)|\nu(\theta)\rangle|^2 \quad (52)$$

$$= -\frac{d-1}{d} < 0. \quad (53)$$

Hence, given $d = 3$, we have shown that we are using a valid witness.

## Witness decomposition

The decomposition of entanglement witness into a series of local measurement is described in refs. 62,63,79. As described in the main text, for a qutrit, a convenient decomposition is in terms of the Gell-Mann matrices. These are defined by the following construction[80],

$$
\begin{aligned}
\chi^{\pm}_{l_e l'_e} = |x^{\pm}_{l_e l'_e}\rangle\langle x^{\pm}_{l_e l'_e}| \quad &\text{and} \quad |x^{\pm}_{l_e l'_e}\rangle = \frac{1}{\sqrt{2}}(|l_e\rangle \pm |l'_e\rangle), \\
\Upsilon^{\pm}_{l_e l'_e} = |y^{\pm}_{l_e l'_e}\rangle\langle y^{\pm}_{l_e l'_e}| \quad &\text{and} \quad |y^{\pm}_{l_e l'_e}\rangle = \frac{1}{\sqrt{2}}(|l_e\rangle \pm i|l'_e\rangle),
\end{aligned}
\quad (54)
$$

the matrices are then given by

$$
\begin{array}{ll}
\lambda_1 = \chi^+_{01} - \chi^-_{01} & \lambda_2 = \Upsilon^+_{01} - \Upsilon^-_{01} \\
\lambda_4 = \chi^+_{0-1} - \chi^-_{0-1} & \lambda_5 = \Upsilon^+_{0-1} - \Upsilon^-_{0-1} \\
\lambda_6 = \chi^+_{1-1} - \chi^-_{1-1} & \lambda_7 = \Upsilon^+_{1-1} - \Upsilon^-_{1-1} \\
\lambda_3 = |0\rangle\langle 0| - |1\rangle\langle 1| & \\
\lambda_8 = \frac{1}{\sqrt{3}}(|0\rangle\langle 0| + |1\rangle\langle 1| - 2|-1\rangle\langle -1|). &
\end{array}
\quad (55)
$$

These matrices can be related to the well-known Pauli matrices. For example, the measurement $\lambda_3 = \hat{L}_{\parallel}$, corresponds directly to an OAM measurement, similar to the $\sigma_z$ Pauli matrix, while $\lambda_1, \lambda_4, \lambda_6$ can be related to $\sigma_x$ as they are all pairwise superposition of two OAM states with the phase $\pm$. The measurements $\lambda_2, \lambda_5, \lambda_7$ relate to $\sigma_y$ as they are pairwise superposition with the phase $\pm i$. Measurements $\lambda_1$–$\lambda_7$ have eigenvalues of $-1, 0, 1$, as with OAM, while $\lambda_8$ has eigenvalues $-2/\sqrt{3}$ and $1/\sqrt{3}$.

The full entanglement witness decomposition given in terms of the Gell-Mann matrices is

$$
\begin{aligned}
\mathcal{W}(\theta) = \frac{1}{12} \Big[ &\frac{8}{3} \mathbb{1}^{\otimes 2} - \lambda_3^{\otimes 2} - \kappa^2(\kappa^{-4}+1)(\lambda_4^{\otimes 2} + \lambda_5^{\otimes 2}) + \lambda_8^{\otimes 2} \\
&+ \sqrt{3}(\lambda_3 \otimes \lambda_8 + \lambda_8 \otimes \lambda_3) + i\kappa^2(\kappa^{-4}-1)(\lambda_4 \otimes \lambda_5 - \lambda_5 \otimes \lambda_4) \\
&- \kappa(\kappa^{-2}+1)(\lambda_1 \otimes \lambda_6 + \lambda_6 \otimes \lambda_1 + \lambda_2 \otimes \lambda_7 + \lambda_7 \otimes \lambda_2) \\
&- i\kappa(\kappa^{-2}-1)(\lambda_1 \otimes \lambda_7 - \lambda_7 \otimes \lambda_1 - \lambda_2 \otimes \lambda_6 + \lambda_6 \otimes \lambda_2) \Big],
\end{aligned}
\quad (56)
$$

where $\kappa = e^{i\theta}$. This decomposition is specific to the Gell-Mann matrices, however, a general procedure may be outlined for any complete set of measurement bases. Thus, if it is simpler to measure in another way, use another entanglement witness or perform this for a higher-dimensional system, the following recipe can be used to achieve the decomposition. For a complete set of one-particle measurements defined by the operators $\lambda_i$ for $i \in [1, d^2]$ and a known entanglement witness $\mathcal{W}$, the decomposition in terms of the local one-particle observables may be written as

$$\mathcal{W} = \sum_{ij} c_{ij} \lambda_i \otimes \lambda_j, \tag{57}$$

where the coefficients $c_{ij}$ can be used to combined with experimental local expectation values to determine $\mathcal{W}$. This sum can be inverted to find $c_{ij}$ via vectorization, i.e., flattening one dimension. We may define

$$\begin{aligned}\mathcal{W}_{\mathrm{v}} &= \mathrm{vec}(\mathcal{W}) \\ &= (\mathcal{W}_{0,0}, \mathcal{W}_{0,1}, ..., \mathcal{W}_{0,d^2-1}, \mathcal{W}_{1,0}, ..., \mathcal{W}_{d^2-1,d^2-1})^T\end{aligned} \tag{58}$$

and $c_{\mathrm{v}} = (c_{0,0}, c_{0,1}, ..., c_{0,d^2-1}, c_{1,0}, ..., c_{d^2-1,d^2-1})^T$. This reduces the $d^2 \times d^2$ matrix operators, $\mathcal{W}$ and $c$, to $d^4$ dimensional vectors. Now we defined a $d^4 \times d^4$ matrix

$$M_{\mathrm{v}} = (\mathrm{vec}(\lambda_0 \otimes \lambda_0), .., \mathrm{vec}(\lambda_{d-1} \otimes \lambda_{d-1})), \tag{59}$$

where each row is the vectorized matrix $\lambda_i \otimes \lambda_j$ for specific values of $i$ and $j$. Now using these definitions we may rewrite (57) as a linear matrix equation $\mathcal{W}_{\mathrm{v}} = Mc_{\mathrm{v}}$, thus we may obtain the coefficients by inverting, such that $c_{\mathrm{v}} = M^{-1}\mathcal{W}_{\mathrm{v}}$. For our finite-dimensional system with $d = 3$, this can be done quickly and straightforwardly.

## Data availability

The data generated for all figures used in this study have been deposited in the Zenodo database and are freely available at https://doi.org/10.5281/zenodo.6610379.

## Code availability

All codes and scripts are available upon request, andrew.maxwell@phys.au.dk.

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

## Acknowledgements

There were many interesting and useful discussion that predicated this study and thus, we would to thank, Prof. Carla Faria, Prof. Sougato Bose, Prof. Alessio Serafini, Prof. Jens Biegert, Prof. Misha Ivanov, Prof. Olga Smirnova and Dr. Emilio Pisanty. A.S.M. acknowledges funding support from the European Union's Horizon 2020 research and innovation program under the Marie Skłodowska-Curie grant agreement SSFI No. 887153. L.B.M. acknowledges support from the Danish Council for Independent Research (Grant Nos. 9040-00001B and 1026-00040B). M.L. and A.S.M. acknowledge support from ERC AdG NOQIA, State Research Agency AEI ('Severo Ochoa' Center of Excellence CEX2019-000910-S) Plan National FIDEUA PID2019-106901GB-I00 project funded by MCIN/ AEI /10.13039/501100011033, FPI, QUANTERA MAQS PCI2019-111828-2 project funded by MCIN/AEI /10.13039/501100011033, Proyectos de I+D+I 'Retos Colaboración' RTC2019-007196-7 project funded by MCIN/AEI /10.13039/501100011033, Fundació Privada Cellex, Fundació Mir-Puig, Generalitat de Catalunya (AGAUR Grant No. 2017 SGR 1341, CERCA program, QuantumCAT U16-011424, co-funded by ERDF Operational Program of Catalonia 2014-2020), EU Horizon 2020 FET-OPEN OPTOLogic (Grant No 899794), and the National Science Centre, Poland (Symfonia Grant No. 2016/20/W/ST4/00314), Marie Skłodowska-Curie grant STREDCH No 101029393, 'La Caixa' Junior Leaders fellowships (ID100010434), and EU Horizon 2020 under Marie Skłodowska-Curie grant agreement No 847648 (LCF/BQ/PI19/11690013, LCF/BQ/PI20/11760031, LCF/BQ/PR20/11770012).).

## Author contributions

A.S.M., L.B.M., and M.L. all contributed to the writing and proofreading of the article. A.S.M. and M.L. conceived the project, while M.L. and L.B.M. jointly managed and supervised it. A.S.M. carried out the simulations and analytical theory.

## Competing interests

The authors declare no competing interests.
