## [Peer Review File · Nature Communications]

Entanglement of Orbital Angular Momentum in Non-Sequential Double IonizationREVIEWER COMMENTS

Reviewer #1 (Remarks to the Author):

Review of “Entanglement of Orbital Angular Momentum in Non-Sequential Double Ionization” By A. S. Maxwell, L. B. Madsen, and M. Lewenstein.

The authors theoretically study the creation of entangled electron ejected from atoms by the non-sequential double ionization process. Specifically, the entanglement in terms of the orbital angular momentum of the electrons is considered. The results are quite interesting, showing that different degrees of entanglement can be achieved depending on the atomic species. The manuscript is also timely in the sense that there is an emerging interest for connecting the fields of quantum information and atomic/molecular physics, and one connection can be the study of entanglement in photoionization processes.

That said, I would have liked to see a bit more of the analysis of the connection between entanglement and (for example) the strong-field dynamics. The authors show that the two electrons are entangled in the orbital angular momentum, but this is almost obvious, since the two electrons are produced from the same atom in a correlated process. What does the value of the entanglement (“logarithmic negativity”) tell us about the recollision process? Or what can we learn about the intermediate state dynamics from measuring an entangled electron pair? In the introduction, the use of entanglement for attosecond imaging is mentioned, but no concrete ideas of how this could be done are presented. In my opinion, this kind of analysis must be presented if the manuscript is to qualify for publication in Nature Communications.

Besides the above comment, I have several additional remarks as detailed below.

1. In the introduction, the authors write “However, all these studies involved the calculation of a continuous density matrix”. This is not true, in Ref. [26] and also [PRA 100, 013421 (2019)], the density matrix for the discrete vibrational quantum numbers is derived.

2. There is a substantial amount of previous research on the properties of entangled photons emitted from atomic systems, see for examples [Phys. Rev. A 77, 022507 (2008)], [Phys. Rev. A 83,032506 (2011)] and the review [J. Phys. B: At. Mol. Opt. Phys. 44 192001 (2011)]. These studies should be mentioned in the introduction. Some discussion could also be made of studies like [Science 318, 949-952 (2007)], where entanglement is experimentally measured in photoionization. I would like to know if

there is anything special in the “strong-field” entanglement as compared to entanglement among particles created by the usual perturbative processes.

3. To quantify the entanglement, the authors are using the “logarithmic negativity”. I would appreciate a brief discussion of this particular entanglement measure compared with others such as the von Neuman entropy, the purity, or the concurrence. What is the physical meaning of the “logarithmic negativity”?

4. Only the RESI type of NSDI is considered in the manuscript. I understand that the creation of a superposition state in the ion before the second ionization event enhances the entanglement. But is the entanglement really zero in the standard, direct-ejection NSDI? Or just small?

5. The Keldysh parameter is taken to be around 1, which is on the border between multiphoton and tunneling. Is the tunneling process itself important for the creation of the entanglement? Differently put, could an entangled two-electron state also be created if the ionization of the second electron proceeds by multiphoton absorption?

Reviewer #2 (Remarks to the Author):

Reviewer's report on the manuscript

"Entanglement of Orbital Angular Momentum in Non-Sequential Double Ionization"

by Andrew S. Maxwell, Lars Bojer Madsen, and Maciej Lewenstein

The authors investigate the quantum entanglement between two photoelectrons generated via the RESI path to NSDI,

based on the innovative idea to utilize electron OAM: to employ electron vortex states in the calculations, which are done in the framework of SFA, and to consider OAM measurements of the photoelectrons.

Simplified treatment of the multielectron atoms (doubly charged ion + 2 electrons), conservation laws valid

at least in dipole approximation, and some additional minor assumptions lead to a surprisingly simple 3×3 electron OAM density matrix, if electron momenta are averaged out or projected to specific values.

The authors then compute, evaluate and analyze the truly interesting and new results of and based on electron pair OAM entanglement,

justifying also that these may give new physical insight into the involved strong-field process.

This novel approach via vertex states circumvents some problems of dealing with continuous variable entanglement,

addressed also in the Introduction, thus it is a valuable advance in the field.

(On the other hand, quite much seems to be disregarded in order to obtain a 3×3 OAM density matrix: entanglement in the averaged degrees of freedom.)

It is also an important merit of the manuscript that it develops the overlap between strong-field physics and quantum information, which

has hidden values, and it will be interesting to both of these research communities.

Whether this manuscript deserves the publication in Nature Communications, depends actually on the assessment

whether the simplifications and assumptions in the model and calculations do not hinder them to capture the essential features of reality correctly,

because a comparison with real experimental results seems not possible now (but hopefully will be in the near future).

I tend to answer this question positively, i.e. I suggest the publication, although I have some concerns:

- the treatment of the rescattering and excitation in the "ion + 2 electrons" model may turn out oversimplified now that entanglement is in the focus, not just momentum or energy,
- the electron spin is not considered at all (this is also noted at the end of the Discussion).

The following minor comments may also contribute to improve or clarify the manuscript:

1. Regarding the 3rd paragraph of Introduction:

Continuous variable entanglement can be analytically calculated in some cases, see e.g. Benedict et al, J. Phys. A: Math. Theor. 45, 085304 (2012).

2. Based on Eqs. (3)-(5) and the text between them, the text before Eq. (3) may be misleading,

since it is not (yet) clear for the reader what does the "two-electron scattering state" mean, please consider rephrasing this sentence.

3. Regarding Figs.3 and 4: it seems better to give the matrix elements' magnitude and phase numerically, than color-coding the magnitude.

4. PPT subsection of Methods section: "(such as our $3 \otimes 2$ system)" should be $3 \otimes 3$, I think.

Reviewer #3 (Remarks to the Author):

This work studies entanglement in nonsequential double ionization (NSDI) process. Using SFA theory, the authors show that entanglement in the orbital angular momentum of two photoelectrons can be generated due to strong correlation between two electrons in rescattering-excitation with subsequent field ionization (RESI) process, which is more prevalent in the low intensity region. Computation of The computed logarithmic negativity reveals that the entanglement is robust to incoherent effects and an entanglement witness is analyzed to minimize the number of measurements in detecting the entanglement in the experiment. A large range of targets and laser parameters is also calculated to find the case with the most entangled electron pairs.

I think this is a very interesting paper since entanglement, which is one of the most important intrinsic features of quantum system, has hardly been addressed in strong-field atomic physics till now. The theoretical work presented in this paper might be a guide for the experiment to detect the entanglement in ultrafast dynamics of atoms and molecules. I would like to recommend publication of the paper after the following points have been addressed by the authors.

1. After great effort made in many years, accurate description of NSDI remains an unsolved problem. Neglect of the ionic Coulomb potential and effect of external field on the bound states are two main shortcomings of the SFA model for RESI, which make the quantitative investigation of the theory on subtle structure in the measurement of NSDI questionable. I think some discussion relevant to the validity of the calculation in study of entanglement is necessary, for example, how will the ionic Coulomb potential influence the result like that proposed in the recent published paper (Hao et al., Commun. Phys. 5, 31 (2022)).

2. Many fringes can be observed in the electron-electron correlation distribution shown in Fig. 3 which is due to interference. It is difficult to understand that these fringes hardly change as shown in Fig. 4 which is obtained after considering focal averaging. It should be noted that no interference fringes have been observed in NSDI experiment so far which is believed to be mainly due to focal averaging effect.

Below we include a point-by-point reply to each of the issues raised by the reviewers.

Reviewer 1:

“The manuscript is also timely in the sense that there is an emerging interest for connecting the fields of quantum information and atomic/molecular physics, and one connection can be the study of entanglement in photoionization processes.” We thank the reviewer for this assessment.

“That said, I would have liked to see a bit more of the analysis of the connection between entanglement and (for example) the strong-field dynamics. The authors show that the two electrons are entangled in the orbital angular momentum, but this is almost obvious, since the two electrons are produced from the same atom in a correlated process. What does the value of the entanglement (“logarithmic negativity”) tell us about the recollision process? Or what can we learn about the intermediate state dynamics from measuring an entangled electron pair? In the introduction, the use of entanglement for attosecond imaging is mentioned, but no concrete ideas of how this could be done are presented.”

We thank the reviewer for this comment, we agree that the logarithmic negativity has the potential to reveal important information regarding the recollision dynamics and the intermediate state. In the revised manuscript (see the updated discussion of Figs. 2 and 3), we now provide specific expressions for the logarithmic negativity and entanglement witness, that relate these quantities directly to OAM transition amplitudes. Through this relation we can clearly relate the entanglement to the intermediate state populations as well as the coherence and phases imparted by the recollision and excitation process. We demonstrate that the entanglement witness provides an observable that can reveal the level of coherence between excitation channels with differing quantum magnetic numbers. On the other hand, the momentum dependent logarithmic negativity directly expresses the phases imparted by the strong field dynamics on each OAM channel.

It is true, the fact the electron are entangled is not so surprising. However, we think that our quantitative assessment of the entanglement is valuable and useful knowledge for the community. Furthermore, the high level of entanglement (maximally entangled for certain final momenta), that is robust to intensity averaging and protected from decoherence with the ion is surprising. We have also added content just before the section “Discussion”, which explores entanglement for attosecond imaging in more depth.

The reviewer also had some additional remarks, which we discuss below:

1. *“In the introduction, the authors write “However, all these studies involved the calculation of a continuous density matrix”. This is not true, in Ref. [26] and also [PRA 100, 013421 (2019)], the density matrix for the discrete vibrational quantum numbers is derived.”*

We thank the reviewer for bringing this to our attention. We now include the additional reference, have changed “all” to “the majority of” and include a qualifying footnote stating “exceptions include studies focused on entanglement involving discrete vibrational states”, citing the above articles.

2. *“There is a substantial amount of previous research on the properties of entangled photons emitted from atomic systems, see for examples [Phys. Rev. A 77, 022507 (2008)], [Phys. Rev. A 83,032506 (2011)] and the review [J. Phys. B: At. Mol. Opt. Phys. 44 192001 (2011)]. These studies should be mentioned in the introduction. Some discussion could also be made of studies like [Science 318, 949-952 (2007)], where entanglement is experimentally measured in photoionization. I would like to know if there is anything special in the “strong-field” entanglement as compared to entanglement among particles created by the usual perturbative processes.”*

We thank the referee again, for bringing these references to our attention. In the amended manuscript we mention the first three studies in the first paragraph. The last reference is cited in the new section that discusses exploiting entanglement for attosecond imaging, just before the section “Discussion”. In strong-field induced entanglement we have the possibility of a laserdriven semi-classical recollision between the electrons. This is what drives the correlation between two electrons, that can be independent of the ion. In an equivalent perturbative process there is no recollision and thus the two electrons will remain entangled to the ion and more vulnerable to decoherence. We now mention this advantage, in the aforementioned section.

3. *“To quantify the entanglement, the authors are using the “logarithmic negativity”. I would appreciate a brief discussion of this particular entanglement measure compared with others such as the von Neuman entropy, the purity, or the concurrence. What is the physical meaning of the “logarithmic negativity”?”*

Now, when we introduce the logarithmic negativity, we give further details on the formulation and physical meaning as well a brief statement comparing to other entanglement measures. We also refer to the new supplementary material, where we introduce the logarithmic negativity in more detail and provide a more in-depth comparison with other entanglement measures.

4. *“Only the RESI type of NSDI is considered in the manuscript. I understand that the creation of a superposition state in the ion before the second ionization event enhances the entanglement. But is the entanglement really zero in the standard, direct-ejection NSDI? Or just small?”*

Our initial analysis suggested that this should be small as the $l=l'=0$ OAM state would dominate. We have now performed further analysis by computing the electron-impact ionization prefactors explicitly in order to estimate the logarithmic negativity. We find in general the above statement to be true, the largest case was for neon, where the logarithmic negativity reached a maximum value of around $E_N=0.5$. This is still much less than the RESI case, where for specific momenta maximally entangled states with $E_N=1.58$ could be achieved. The details of this calculation are included in the new supplementary material. We have added a short comment in main manuscript to refer to this just before the section “Entanglement measure and witness”.

5. *“The Keldysh parameter is taken to be around 1, which is on the border between multiphoton and tunneling. Is the tunneling process itself important for the creation of the entanglement?—Differently put, could an entangled two-electron state also be created if the ionization of the first second electron proceeds by multiphoton absorption?”*

We thank the referee for the interesting question. In this case the requirement upon the tunnelling steps is only that the orbital angular momentum (OAM) is conserved. This would also occur in multiphoton ionization for the same field configuration. What is crucial is the recollision to enforce OAM correlation and the existence of intermediate states which lead to the same final state to ensure the formation of coherent superposition. We have added a footnote stating this on page 6.

Reviewer 2:

“This novel approach via vertex states circumvents some problems of dealing with continuous variable entanglement, addressed also in the Introduction, thus it is a valuable advance in the field.”

“It is also an important merit of the manuscript that it develops the overlap between strong-field physics and quantum information, which has hidden values, and it will be interesting to both of these research communities.”

We thank the referee for these remarks.

“Whether this manuscript deserves the publication in Nature Communications, depends actually on the assessment whether the simplifications and assumptions in the model and calculations do not hinder them to capture the essential features of reality correctly, because a comparison with real experimental results seems not possible now (but hopefully will be in the near future). I tend to answer this question positively, i.e. I suggest the publication, although I have some concerns:

- the treatment of the rescattering and excitation in the "ion + 2 electrons" model may turn out oversimplified now that entanglement is in the focus, not just momentum or energy,*
- the electron spin is not considered at all (this is also noted at the end of the Discussion).”*

We agree with the reviewer that entanglement in the context of strong-field physics is somewhat uncharted territory and requires careful consideration. For larger atomic targets such as argon the additional valence electrons could play a role. However, for the smaller group 2 elements, particularly beryllium, the ion+2 electrons model is likely to be well-justified. For beryllium the two remaining core electron are tightly bound in the filled 1s shell so are unlikely to play a role. In order to fully justify this would require a many-electron (>2) model, which is well beyond current capabilities. We can however use the energy required to excite a third electron, from the doubly charged ion as a guide. For beryllium the lowest possible transition for a core electron (1s→2s) has an energy of 4.3 a.u. and for magnesium (2p→ 3s) has an energy of 1.9 a.u., which are both far too high to be accessible by this process. In the case of argon the transition (3s²3p⁴→3s3p⁵) with an energy of 0.37 a.u suggests that a multielectron treatment may play a role. Multielectron effects could add alternative OAM channels associated with different final ionic states, that will contribute incoherently and thus reduce the overall entanglement. However, given that our main focus is on beryllium and magnesium, where, the ion+2 electrons model should work the best, we have confidence in our main results and conclusions. We can also rule out coupling to the external OAM of the ion given its much larger mass. We have added a summary of this to the second paragraph of the section “Measurement Considerations”.

With respect to spin, we agree it is an interesting and important consideration. However, we do consider certain spin related effects already. The primary effects of spin will be related to the initial state. In this case we only need to consider spin singlet states, thus resulting in symmetric symmetrization of the remainder of the wave function, which we already employ. Furthermore, there

will be spin polarization effects through the fine structure splitting of the intermediate excited state. However, this energy splitting is small for the systems in question less than 1% for argon and less than 0.03% for beryllium. Thus, in these cases it can safely be neglected. The spin-orbit coupling in strongfield dynamics, such as in the recollision, has been neglected as is currently wide-spread in the strongfield community. We appreciate we did not include all these details in the original manuscript and have added a summary of the above to the section "Discussion".

The reviewer also had some additional remarks, which we discuss below:

1. *"Regarding the 3rd paragraph of Introduction: Continuous variable entanglement can be analytically calculated in some cases, see e.g. Benedict et al, J. Phys. A: Math. Theor. 45, 085304 (2012)."*

We thank the referee for bringing this to our attention and in the revised version of the manuscript we cite this work and mention that in certain cases an analytic treatment is possible.

2. *"Based on Eqs. (3)-(5) and the text between them, the text before Eq. (3) may be misleading, since it is not (yet) clear for the reader what does the "two-electron scattering state" mean, please consider rephrasing this sentence."*

We thank the referee for pointing this out, we now use the term "the final state of the two electrons in the continuum".

3. *"Regarding Figs.3 and 4: it seems better to give the matrix elements' magnitude and phase numerically, than color-coding the magnitude."*

In the revised manuscript we have changed figures 3 and 4 so that the density matrices display the phase and magnitude as text, and we have removed the colour bar.

4. *"PPT subsection of Methods section: "(such as our 3 \otimes system)" should be 3 \otimes , I think."*

Thank you, we have corrected this typo.

Reviewer 3:

"I think this is a very interesting paper since entanglement, which is one of the most important intrinsic features of quantum system, has hardly been addressed in strong-field atomic physics till now. The theoretical work presented in this paper might be a guide for the experiment to detect the entanglement in ultrafast dynamics of atoms and molecules. I would like to recommend publication of the paper after the following points have been addressed by the authors." We thank the referee for this positive assessment.

The referee suggested two points to be addressed:

1. *"After great effort made in many years, accurate description of NSDI remains an unsolved problem. Neglect of the ionic Coulomb potential and effect of external field on the bound states*

ff are two main shortcomings of the SFA model for RESI, which make the quantitative investigation of the theory on subtle structure in the measurement of NSDI questionable. I think some discussion relevant to the validity of the calculation in study of entanglement is necessary, for example, how will the ionic Coulomb potential influence the result like that proposed in the recent published paper (Hao et al., Commun. Phys. 5, 31 (2022))."

We agree with the referees comments, this is an important consideration. The recent paper, which provides an improved agreement with experimental work in New J. Phys. 16 (2014) 033008 (Ref. [14] of supplementary material), employs a Monte Carlo treatment for the first electron and the CCSFA for the second electron ionization and propagation. The recollision and excitation step uses the same S-matrix formalism, this is key, as it will lead to exactly the same superposition across OAM values, thus entanglement via the same mechanism will occur. One of the main novel features identified by the authors in this work is importance of recollision of the second electron after ionization from the excited state. This will change the shape of the momentum distribution but will not affect the final OAM and thus the entanglement should be of the same magnitude. In fact, such trajectories would allow holographic measurements of a target and reinforces the idea of entanglement-enhanced holography. Furthermore, the inclusion of these additional trajectories may lead to some interferences currently visible in this SFA formalism to become washed out or less clear. In the revised manuscript in the second paragraph after the section "Measurement Considerations", we comment on the validity of the approach, including the effect of the Coulomb potential and cite the suggested reference. We also discuss how the entanglement could be used to enhance holographic measurements in the section just before "Discussion".

2. *"Many fringes can be observed in the electron-electron correlation distribution shown in Fig. 3 which is due to interference. It is difficult to understand that these fringes hardly change as ffi shown in Fig. 4 which is obtained after considering focal averaging. It should be noted that no interference fringes have been observed in NSDI experiment so far which is believed to be mainly due to focal averaging e ect."*

This is an important point raised by the reviewer. The interferences can be broken down into two categories, "channel interferences" arising from the superposition of intermediate states and "event interferences" arising both from ionization at differing times in the laser pulse and symmetrization to account for the indistinguishable electrons. Channel interference leads to broad fringes that affect the overall shape of the momentum distribution and were proposed for forming the correct distribution shapes for short pulses, Phys. Rev. Lett. 112, 1273 073002 (2014) and Phys. Rev. Lett. 116, 1280 143001 (2016) ([69] and [71] in manuscript). Event interference, leads to the fine fringes pointed out by the reviewer and are not directly related to the OAM entanglement. This can be shown by switching on/ off event interference, see for example Fig. 10 in Phys. Rev. A 92, 023421 (2015) ([70] in manuscript).

Experiments that are restricted to address the RESI mechanism have a lower intensity, and thus, electron statistics are lower making it difficult to collect high resolution data that clearly

show the fine interferences. A particularly useful benchmark has been *New J. Phys.* 16 (2014) 033008 (Ref. [14] of supplementary material), used by many theoretical models. Here, it can be seen the resolution is not yet at a level to clearly observe the fine event interference. However, subsequent experimental work has looked at oscillations in an asymmetry parameter over different laser intensities and argued this can only be explained by such interferences in *Phys. Rev. A* 96, 032511 (2017) ([72] in manuscript).

Some additional considerations are that the inclusion of the Coulomb potential, such as in *Hao et al., Commun. Phys.* 5, 31 (2022), would lead to the addition of recolliding electron trajectories for the second electron. This would mean the sharp SFA interferences seen in the single electron distribution of the second electron, would become the subtler (but more useful) holographic interferences. It is more likely that these would be washed out in the two electron correlated momentum distribution after integration over perpendicular momentum.

In the manuscript we now comment on the fringes mentioned by the referee and summarize the key features of the above discussion. We do this in the discussion of Fig. 4 (in the original manuscript), which has been moved to page 2 of the supplementary material.

Yours Sincerely,

Andrew S. Maxwell, Lars Bojer Madsen, Maciej Lewenstein

REVIEWERS' COMMENTS

Reviewer #1 (Remarks to the Author):

I think that the authors have made a good job in revising the manuscript in line with my (and the other referee's) comments.

One may hope that the theoretical proposal in the paper will lead to an interest also on the experimental side for the measurement of entanglement in strong-field processes.

I recommend publication of the manuscript in its present form.

Reviewer #2 (Remarks to the Author):

The authors addressed all of my comments, most of them convincingly, and they made the necessary changes to the manuscript accordingly.

The new section on Entanglement-Enhanced attosecond imaging, triggered by Reviewer 1, is a valuable addition to the content.

I recommend publication of the manuscript.

Reviewer #3 (Remarks to the Author):

I appreciate the author's effort to improve the manuscript and my first concern has been well addressed. My second concern and the discussion between the authors and other reviewers indicate that the experimental observation of the entanglement in NSDI process might be difficult since the model is oversimplified (for example, the EI process, which has much lower E_N , will contribute to the electron-electron correlation distribution) and the effect is much stronger for alkaline earth atoms, which has never been measured using COLTRIMS, than for rare gas atoms. However, considering that this is the first theoretical attempt to deal with a novel problem which has broad interests, I recommend publication of this paper in Nat. Communication. Finally, I suggest the authors to add some more discussions on the difficulties in experiment on behalf of the readers, especially from the experimental side.

Reviewer 1: *“I think that the authors have made a good job in revising the manuscript in line with my (and the other referee's) comments. One may hope that the theoretical proposal in the paper will lead to an interest also on the experimental side for the measurement of entanglement in strong-field processes.*

I recommend publication of the manuscript in its present form.”

We thank the reviewer for their helpful and constructive comments and echo the sentiment regarding experiments.

Reviewer 2: *“The authors addressed all of my comments, most of them convincingly, and they made the necessary changes to the manuscript accordingly. The new section on Entanglement-Enhanced attosecond imaging, triggered by Reviewer 1, is a valuable addition to the content. I recommend publication of the manuscript.”*

We thank the reviewer for their helpful and constructive comments.

Reviewer 3: *“I appreciate the author's effort to improve the manuscript and my first concern has been well addressed. My second concern and the discussion between the authors and other reviewers indicate that the experimental observation of the entanglement in NSDI process might be difficult since the model is oversimplified (for example, the EI process, which has much lower E_N , will contribute to the electron-electron correlation distribution) and the effect is much stronger for alkaline earth atoms, which has never been measured using COLTRIMS, than for rare gas atoms. However, considering that this is the first theoretical attempt to deal with a novel problem which has broad interests, I recommend publication of this paper in Nat. Communication. Finally, I suggest the authors to add some more discussions on the difficulties in experiment on behalf of the readers, especially from the experimental side.”*

We thank the reviewer for their helpful and constructive comments. We agree that there are certain hurdles in realizing this in experiment. Regarding the EI process, there will be parameter ranges where this can be minimized and safely neglected. However, in future the best approach will be to model both processes directly. Although there have been studies on NSDI for alkaline earth metals, it is true for COLTRIMS this may be a difficult task. An alternative route is to consider a broader parameter range or even tailored fields, with which we may boost the entanglement of the rare gases or other more accessible targets. We now provide a similar discussion on experimental difficulties in the “Discussion” section.

Yours Sincerely,

Andrew S. Maxwell, Lars Bojer Madsen, Maciej Lewenstein